# Towards Real-Time BCI for Speech with Whisper-Based Decoding of Neural Activity

## Abstract

Decoding continuous speech from neural activity is a central challenge for brain–computer interfaces (BCIs), with major implications for restoring communication in individuals with paralysis. While recent work has achieved impressive performance using recurrent neural decoders trained on multi-electrode array (MEA) recordings, these models remain brittle, data-hungry, and struggle to generalize across sessions or participants. In this work, we introduce Whisper-BCI, the first neural speech decoder to integrate high-resolution MEA recordings with a large pretrained automatic speech recognition (ASR) model. Our approach leverages interpretability findings showing that Whisper's encoder layers learn phoneme-selective representations with localized attention. Building on this insight, we adapt Whisper to predict phoneme embeddings from neural signals into the third layer of Whisper's encoder and fine-tune the model end-to-end with a hybrid objective combining CTC loss on phoneme alignments and cross-entropy loss on word tokens. We further introduce domain-informed modifications including windowed self-attention to capture articulatory continuity, day-specific low-rank projections to address non-stationarity and reduce computational complexity, and subject-specific input embedders for cross-subject training. Evaluated on Card et al. and Brain-to-Text '25 data, Whisper-BCI performs on par with or outperforms baselines relative to prior MEA decoders, and achieves cross-subject generalization, opening the door to robust decoding with limited resources. Post-processing with rescoring and grammar-guided correction yields an additional relative improvement, and the use of windowed attention has the potential to significantly reduce latency, enabling near-real-time online decoding. Our results demonstrate that pretrained ASR models can serve as effective language backbones for neural decoding and suggest a scalable path toward foundation models for speech BCIs.

## 1 Introduction

The ability to restore fluent communication to individuals with severe speech impairments is one of the most compelling goals of brain–computer interface (BCI) research. Conditions such as brainstem stroke, ALS, or locked-in syndrome can sever the neural-to-muscle pathways necessary for speech, leaving affected individuals unable to communicate despite preserved cognition. Neural speech decoding aims to bypass this damaged periphery by directly reading out neural activity from speech-related cortical regions and translating it into text or synthesized voice in real time (Silva et al., 2024).

Two main technological directions have been explored for neural speech decoding: non-invasive approaches such as EEG, MEG, or fMRI, and invasive approaches such as electrocorticography (ECoG) and penetrating microelectrode arrays (MEAs). Although non-invasive approaches have demonstrated promising results in limited settings, invasive systems currently provide the temporal and spatial resolution necessary for natural speech restoration, with recent studies achieving vocabularies of thousands of words and word error rates (WERs) below 20% (Willett et al., 2023; Moses et al., 2021; Card et al., 2024).

Much of this progress has been driven by the Willett et al. (2023) decoder, which has become the de facto baseline in the field. Their two-stage pipeline first employs a GRU trained with a Connectionist Temporal Classification (CTC) loss (Graves et al., 2006) to predict phonemes, followed by a

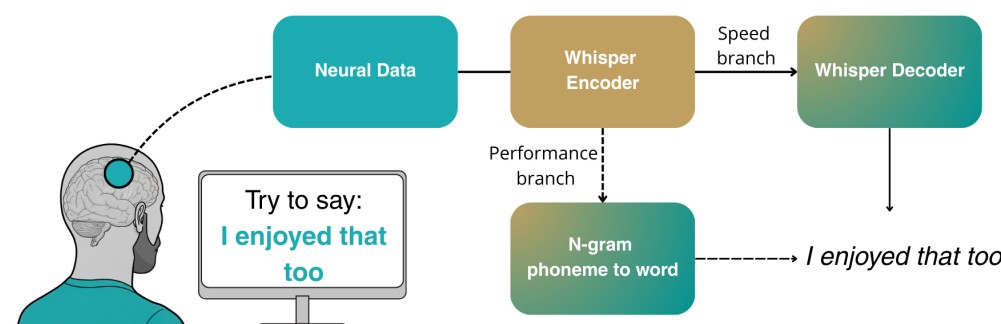

Figure 1: Proposed Whisper-based neural decoder. Neural data are transformed and embedded before being fed into a modified Whisper encoder. Two complementary decoding paths are supported: a *performance branch* that predicts phonemes and uses an external $n$-gram WFST for high-accuracy decoding, and a *speed branch* that directly generates text through Whisper's decoder for low-latency, end-to-end decoding.

powerful language-modeling stage that combines an $n$-gram decoder with rescoring from both a second $n$-gram and a large language model (LLM). Subsequent studies have extended this architecture in different directions. For instance, ensembling strategies combined with a transcription selection mechanism based on in-context learning (LISA) were introduced in (Benster et al., 2024), which reports improved WERs on benchmark datasets. Li et al. (2024) proposed replacing phonemes with diphones as intermediate targets, showing that context-aware subphonemic units can improve modeling power. Their method, when combined with an enhanced LISA-style selection stage that leveraged GPT-3.5, achieved a state-of-the-art WER of 5.77% in the Brain-to-Text '24 competition. In parallel, transformer-based approaches have also been explored: Feghhi et al. (2025) introduced a compact transformer architecture that surpassed the Willett baseline while using substantially fewer parameters, though still falling short of the strongest RNN-based systems. End-to-end strategies have also been attempted: Feng et al. (2024) coupled neural feature extractors to LLMs (e.g., Llama 2 (Touvron et al., 2023)), though in practice these systems did not outperform the Willett baseline. Most recently, Card et al. (2024) applied optimized Willett-style decoders to multi-unit array recordings, achieving the lowest WERs reported to date on both prompted and unstructured conversational speech.

Beyond text decoding, alternative lines of work have investigated direct speech synthesis from neural signals (Metzger et al., 2023; Littlejohn et al., 2025; Wairagkar et al., 2025). These systems typically predict intermediate acoustic or linguistic representations such as HuBERT units (Hsu et al., 2021) or spectral–pitch features, which are then passed to vocoders to produce audible speech. While such approaches enable real-time intelligible synthesis, their accuracy still lags behind text-based decoding pipelines.

Despite these advances, several critical challenges remain unsolved. Neural recordings are highly non-stationary, drifting across sessions and days, which leads to rapid degradation in performance unless models are retrained or aggressively regularized. Data scarcity also remains a major issue: typical datasets comprise on the order of $10^4$ sentences per participant, a scale at which overfitting is a constant risk. Finally, most prior work has focused on single-participant decoders, leaving open the question of how to build generalizable systems that operate robustly across individuals.

To address these limitations, researchers have increasingly drawn inspiration from automatic speech recognition (ASR), where large-scale pretraining and robust representation learning have revolutionized performance. For example, Yang et al. (2024) recently proposed NeuSpeech, a variant of Whisper (Radford et al., 2022) fine-tuned on MEG recordings, suggesting that pre-trained ASR backbones can serve as promising bridges between neural and acoustic domains. However, while effective, these systems remain computationally demanding, limiting their applicability in online settings.

In this work, we take a different direction and introduce a neural speech decoder[1] that leverages the representational power of large, pretrained ASR models. We build on Whisper, a state-of-the-art multilingual ASR system trained on 680k hours of speech, whose encoder has been shown through recent interpretability studies to learn phoneme-selective features in its early layers and to operate effectively with highly localized attention patterns (Reid, 2023). These findings suggest that Whisper can act as a powerful phonetic backbone for neural decoding, provided that neural data can be projected into its representational space.

Our approach aggregates all publicly available microelectrode-array datasets for speech decoding, including those in (Willett et al., 2021) and (Card et al., 2024), to train a unified Whisper-inspired model capable of joint phoneme and token prediction (see Figure 1). The model supports two complementary decoding paths: one that predicts phonemes directly and can be coupled with a conventional WFST decoder and rescoring pipeline for maximum accuracy, and another that leverages Whisper's decoder as a weak language model, enabling near-instant decoding via greedy or small beam-search. This dual design allows us to trade off accuracy and latency depending on application constraints, making the system suitable both for offline evaluation and for low-latency online BCI use.

To mitigate the deleterious effects of session-to-session and subject-to-subject variability, we introduce a hierarchical normalization scheme that first applies a global month-level projection capturing slow population-level shifts and then applies a low-rank day-specific refinement that accounts for finer-grained variability. This approach regularizes day transforms by enforcing smoothness across temporally adjacent sessions and improves generalization across subjects and recording days. Combined with windowed self-attention in the early encoder layers, which incorporates articulatory context while reducing computational cost, this design yields a robust, data-efficient decoder that makes significant progress toward real-time neural speech decoding.

## 2 METHODS

### 2.1 DATASETS

To train and evaluate our proposed decoder, we assembled what is, to our knowledge, the most comprehensive collection of publicly available intracortical neural recordings for speech decoding. Our corpus combines data from two major sources within the BrainGate2 clinical trial, thereby capturing a wide range of sessions, recording conditions, and cortical sampling locations.

The first dataset, which we refer to as **Willett**, consists of recordings from participant T12 (Willett et al., 2023), an individual with amyotrophic lateral sclerosis (ALS) and complete anarthria. This participant was chronically implanted with four 64-channel Utah microelectrode arrays (256 channels total) targeting two functionally critical nodes of the speech network: the ventral premotor cortex (area 6v) and Broca's area (area 44). Implant sites were selected on the basis of subject-specific fMRI activations during attempted speech and cortical parcellations derived from the Human Connectome Project. During each session, the participant was cued to attempt production of prompted sentences in an instructed-delay task. Raw signals were band-pass filtered in the high-gamma band (250–5000 Hz), and two neural features were extracted per channel per 20 ms window: (i) threshold crossing counts, detected at $-4.5$ RMS, and (ii) spike-band power, computed as the mean squared voltage in the same window. These features were concatenated to form a channel-level neural representation at 50Hz. The Willett dataset spans 24 days of recordings and approximately 9,000 sentence trials. In our experiments we adopt the official training, test, and competition splits, which are defined over disjoint temporal blocks to assess cross-day generalization. Following prior work, we use only electrodes from premotor cortex, as those in Broca's area contain comparatively little speech-related information.

The second source of data, denoted **Card**, contains recordings from participant T15 (Card et al., 2024), an individual with ALS and severe dysarthria implanted with four Utah arrays (256 channels) in the left ventral precentral gyrus. Neural feature extraction follows the same preprocessing pipeline as in Willett et al., yielding spike counts and spike-band power binned at 20 ms resolution. This dataset covers more than eight months of recordings, distributed over 84 sessions across 45 recording

---

[1]The code we developed is available in the submission attachments.

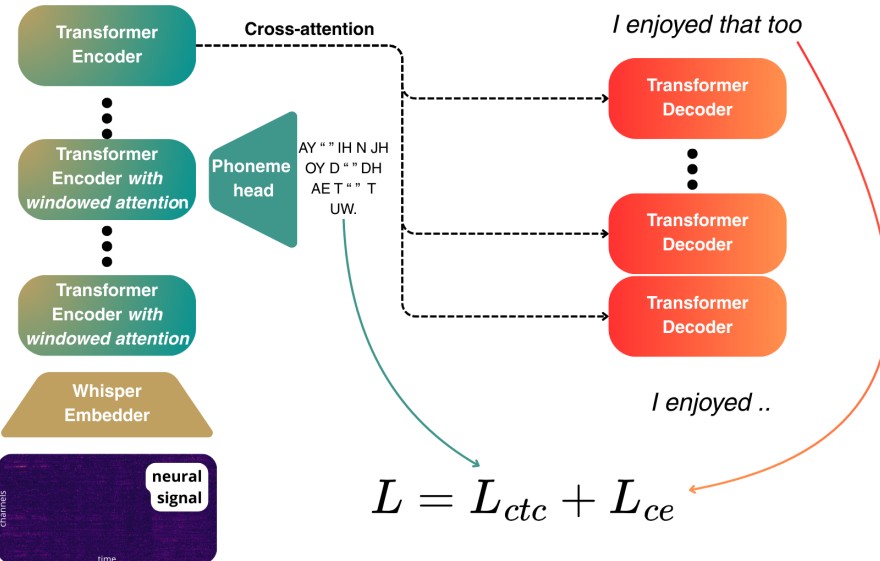

Figure 2: Architecture of the proposed Whisper-based neural decoder. Neural features are first mapped into a shared embedding space using a convolutional front-end (Whisper embedder) and processed by a stack of Transformer encoder layers, with the first $L_p$ layers employing non-causal windowed attention to capture local articulatory context efficiently. A phoneme head attached at layer $L_p$ predicts phoneme sequences using a CTC loss, while the remaining encoder layers feed a Transformer decoder trained with cross-entropy on text tokens. The combined objective $\mathcal{L} = \mathcal{L}_{\text{ctc}} + \mathcal{L}_{\text{ce}}$ encourages the encoder to learn representations that are simultaneously phoneme-discriminative and language-aware.

days, and includes a diverse set of lexical prompts designed to elicit a large phoneme inventory. We again use the official splits provided by the authors to ensure comparability with previously reported results.

By merging these datasets, we create a unified training corpus that maximizes diversity across cortical regions, recording days, and session conditions. This aggregation is crucial for training a Whisper-inspired architecture, which benefits from large, heterogeneous datasets to learn robust neural-to-phoneme mappings. The combined corpus provides a unique opportunity to evaluate cross-subject generalization and to investigate whether a single decoder can approach state-of-the-art performance across participants while maintaining robustness to day-to-day variability. In all cases we used the original train/test/competition splits provided by the authors, collecting 16872 training trials (8072 from (Card et al., 2024), 8800 from (Willett et al., 2023)), 1426 test and 1450 competition trials for Card dataset and 880 test and 1200 competition trials for Willett one.

## 2.2 MODEL

Our proposed decoder adapts the Whisper architecture to neural data, combining phoneme-level supervision with weak language modeling to achieve both high accuracy and low-latency decoding. The model consists of three main components: (i) a neural feature embedder with hierarchical day normalization, (ii) a modified Whisper encoder with windowed self-attention, and (iii) dual decoding heads for phoneme-level and token-level prediction. A schematic overview is shown in Figure 2.

**Neural Embedding and Hierarchical Day Transforms.** Let $\mathbf{X} \in \mathbb{R}^{B \times T \times D}$ denote the neural features for a batch of $B$ trials, with $T$ time bins and $D$ channels (spike counts and spike-band power concatenated along the channel dimension). We first project $\mathbf{X}$ into a subject- and session-invariant space using a hierarchical normalization scheme. Specifically, for each day $d$ we apply a linear transformation

$$\tilde{\mathbf{X}}_{b,t} = \sigma\big[(\mathbf{M}_{m(d)} + \Delta\mathbf{W}_d)\,\mathbf{X}_{b,t} + \mathbf{b}_{m(d)} + \Delta\mathbf{b}_d\big],$$

where $\mathbf{M}_{m(d)} \in \mathbb{R}^{D \times D}$ and $\mathbf{b}_{m(d)} \in \mathbb{R}^D$ are month-level parameters shared across all days in month $m(d)$, $\Delta \mathbf{W}_d = \mathbf{A}_d \mathbf{B}_d$ is a low-rank ($r \ll D$) day-specific correction with parameters $\mathbf{A}_d \in \mathbb{R}^{D \times r}, \mathbf{B}_d \in \mathbb{R}^{r \times D}$, and $\Delta \mathbf{b}_d \in \mathbb{R}^D$ is a day-specific bias. This factorization encourages smoothness across days and prevents overfitting to single-day idiosyncrasies. The nonlinearity $\sigma(\cdot)$ is a softsign activation, which empirically stabilizes training compared to $\tanh$.

The transformed features are then passed through two 1-D convolutional layers with kernel sizes $k_1, k_2$ and strides 1 and $s$:

$$\mathbf{H} = \text{Conv}_2\big(\text{GELU}(\text{Conv}_1(\tilde{\mathbf{X}}))\big),$$

producing a token sequence $\mathbf{H} \in \mathbb{R}^{B \times T' \times d_{\text{model}}}$ of length $T' \approx \lfloor T/s \rfloor$. Sinusoidal day encodings are added to $\mathbf{H}$ to explicitly signal session identity and further assist the model in compensating for distribution shifts across recording days.

**Modified Whisper Encoder with Windowed Attention.** The embedded sequence $\mathbf{H}$ is then fed into a modified Whisper encoder, initialized from the pretrained original checkpoint *whisper-tiny*. Whisper's encoder is a stack of $L$ Transformer blocks, each consisting of multi-head self-attention and feed-forward sublayers. Following interpretability analyses of Whisper (Reid, 2023), which revealed that early encoder layers learn phoneme-selective features with strongly localized receptive fields, we replace the global attention in the first $L_p$ layers with a non-causal windowed attention mechanism. For each token at position $t$, attention is restricted to a symmetric temporal window $\{t - w, \dots, t + w\}$:

$$\text{Attn} = \text{softmax}\left(\frac{\mathbf{Q}\mathbf{K}^T}{\sqrt{d_k}} + \mathbf{M}\right) \mathbf{V},$$

where $\mathbf{M}$[2] is an additive mask set to $-\infty$ outside the allowed window. This choice encodes the inductive bias that articulatory dynamics are locally constrained while reducing computational complexity from $\mathcal{O}(T'^2)$ to $\mathcal{O}(T'w)$ in the early layers, improving latency during online decoding.[3]

**Phoneme Prediction and CTC Loss.** To encourage the encoder to learn a phoneme-aligned representation, we attach a linear prediction head on top of the encoder output of layer $L_p$. To reduce the number of phoneme predictions, we employed a further contextual window of 80ms, aggregating consecutive $q = 2$ token embeddings via unfolding before projecting to class logits:[4]

$$\mathbf{Z}_{b,\tau} = \mathbf{W}_p \, \text{concat}\big[\mathbf{H}_{b,\tau q:(\tau+1)q}\big] + \mathbf{c}_p, \qquad \mathcal{L}_{\text{CTC}} = \text{CTC}\big(\mathbf{Z}, \mathbf{y}_{\text{phoneme}}\big).$$

The CTC loss $\mathcal{L}_{\text{CTC}}$ aligns the predicted phoneme sequence with the ground-truth sequence $\mathbf{y}_{\text{phoneme}}$ without requiring explicit frame-level alignment.

**Token-Level Loss and Weak Language Supervision.** The remaining encoder layers and Whisper's decoder are trained to predict text tokens using the standard cross-entropy objective:

$$\mathcal{L}_{\text{CE}} = -\sum_{b,t} \log p_\theta(y_{b,t} \mid y_{b,<t}, \mathbf{H}_b),$$

where $y_{b,t}$ are text tokens and $\mathbf{H}_b$ is the encoder output for trial $b$. This weak language supervision injects distributional knowledge about word sequences and semantic constraints, enabling the decoder to disambiguate noisy phoneme predictions and to produce fluent transcriptions.

**Decoding Strategies.** At inference time, we support two complementary decoding paths. The first is a phoneme-based path, where $\mathbf{Z}$ is decoded into phoneme sequences using beam search with a weighted finite-state transducer (WFST) language model and optional n-gram rescoring (Willett et al., 2023; Card et al., 2024), maximizing transcription accuracy. The second path bypasses the WFST and performs direct token generation with Whisper's decoder, optionally using a (small) beam search. This path provides near-instant decoding suitable for online BCI applications, with substantially reduced computational cost compared to large-beam phoneme search.

---

[2]In this context, the $\mathbf{M}$ notation does not refer to the aforementioned month-specific transformation.

[3]In order to make this advantage effective, attention layers have to be properly implemented.

[4]Here, the notation : has to be understood as a Python-like interval (right endpoint excluded).

**Training Objective.** The overall training loss is a simple linear combination of the phoneme-level and token-level objectives:

$$\mathcal{L} = \lambda_{\mathrm{CTC}}\, \mathcal{L}_{\mathrm{CTC}} + \lambda_{\mathrm{CE}}\, \mathcal{L}_{\mathrm{CE}},$$

with $\lambda_{\mathrm{CTC}} = \lambda_{\mathrm{CE}} = 1.0$ in our experiments. This joint objective encourages the encoder to produce representations that are simultaneously discriminative for phoneme recognition and predictive for text generation, improving generalization across recording days and subjects.

## 2.3 EVALUATION

We evaluate our models using two complementary metrics that capture performance at the phoneme and word levels. **Phoneme Error Rate (PER)** is defined as the Levenshtein edit distance between the predicted and ground-truth phoneme sequences, normalized by the length of the reference. This metric accounts for substitutions, insertions, and deletions, and provides a direct measure of the quality of the neural encoder's phoneme representations independent of language modeling effects.

**Word Error Rate (WER)** is computed on the final decoded text output, allowing for a direct comparison between WFST-based decoding and neural sequence-to-sequence decoding. WER reflects the end-to-end transcription quality at the sentence level and is particularly relevant for communication BCIs where word-level intelligibility is the primary objective.

All results are reported on the official held-out test/competition splits of each dataset to ensure comparability with previously published work. For the BTT competition setting, where ground-truth phoneme annotations are not provided, we report only WER. Competition WER is obtained using the official online evaluation platform, which provides a standardized and blinded assessment of model generalization on unseen data.

## 3 RESULTS

Table 1: Accuracy in terms of PER and WER for the evaluated models. Results are shown for both the test and competition splits of the Card dataset. The *5-gram* subscript indicates that extensive post-processing with a 5-gram model and rescoring was applied. The *0/1-gram* notation refers either to greedy text generation from the word head in our models (0-gram) or to unigram decoding with rescoring in the baseline (1-gram).

| Model | Test | | | Competition | |
|---|---|---|---|---|---|
| | PER | $\mathrm{WER}_{\text{5-gram}}$ | $\mathrm{WER}_{\text{0/1-gram}}$ | $\mathrm{WER}_{\text{5-gram}}$ | $\mathrm{WER}_{\text{0/1-gram}}$ |
| Baseline | **10.20%** | 7.40% | 21.14% | **6.70%** | 20.91% |
| Ours | 12.78% | 7.88% | 13.85% | 8.76% | 13.85% |
| Ours (cross) | 11.02% | **6.49%** | **11.36%** | **6.70%** | **10.85%** |

Table 2: Computational profile of the evaluated models. For each model, we report approximate RAM usage, VRAM usage, inference time per sentence, number of training steps, and total training time.

| Model | Memory | | Inference time |
|---|---|---|---|
| | RAM | VRAM | |
| Baseline (1-gram) | - | ∼14GB | ∼0.75s |
| Baseline (5-gram) | ∼300GB | ∼14GB | ∼0.75s |
| Ours (0-gram) | - | ∼2GB | ∼0.05s |
| Ours (5-gram) | ∼300GB | ∼14GB | ∼0.75s |

| | Training steps | Training time |
|---|---|---|
| Baseline | 120k | ∼6.5h |
| Ours | 20k | ∼2.5h |
| Ours (cross) | 26.4k | ∼3.5h |

| Target: | Put that back in the saucepan. |
| Transcription (5-gram): | Put that back in the **sustain**. |
| Transcription (0-gram): | Put that back in the **sustain**. |
| | |
| Target: | I just stood there and soaked it all up. |
| Transcription (5-gram): | I just stood there and soaked it all up. |
| Transcription (0-gram): | I just **start** there and **start** it all up. |
| | |
| Target: | Not too controversial. |
| Transcription (5-gram): | Not too controversial. |
| Transcription (0-gram): | Not too **extrafacial**. |

Figure 3: Examples of decoded sentences alongside their target references. Transcriptions with (5-gram) and without (0-gram) post-processing are shown. Sentences have been selected to highlight errors, illustrating the types of mistakes made by the models.

We evaluate our proposed model under two training regimes: trained solely on the Card dataset, and in a cross-dataset setting, jointly on Card and Willett (indicated as *"(cross)"* in the tables). As a baseline, we use the Willett-style decoder introduced in (Card et al., 2024), trained exclusively on Card following the original procedure. Details of our training configurations are provided in Appendix A. All models are evaluated only on Card data, specifically on the official test split released with the dataset, and the competition split from the Brain-to-Text '25 challenge,[5] which pools trials from the same acquisitions (possibly non-overlapping).

Table 1 reports results in terms of PER and WER across both splits. For WER, we distinguish between outputs with post-processing (5-gram WFST decoding with OPT-based rescoring, denoted *"5-gram"*) and those without. In the latter case, the baseline uses unigram decoding plus rescoring (the minimal pipeline needed to convert phoneme predictions into intelligible sentences, denoted *"1-gram"*), while our models decode greedily from the word head (*"0-gram"*).

The relative ranking of models differs depending on the metric. On the test split, the baseline achieves the lowest PER, confirming its strength as a phoneme-level model. However, what ultimately matters for practical communication is how decoding propagates phoneme errors into words. Here, our model–especially the cross-dataset variant–achieves substantially lower WERs. On the competition split, PER is unavailable since ground-truth phoneme labels are not provided; WERs are obtained via submission to the Kaggle platform. In this case, under 5-gram decoding our model performs on par with the baseline, while under 0/1-gram decoding it achieves significantly lower error rates, highlighting the robustness of our approach in settings without heavy post-processing.

Table 2 provides complementary results on resource requirements. Both the baseline and our models require ∼2GB of VRAM to produce logits. When OPT-based rescoring is used, an additional ∼12GB of VRAM is needed, and 5-gram decoding further requires ∼300GB of RAM. In terms of inference latency, performance is dominated by the language modeling stage whenever n-gram decoding is involved: ∼0.75s per sentence, compared to ∼0.05s for greedy decoding with our model. Training efficiency also favors our approach. Despite being trained on more data in the cross-dataset regime, our model converges in roughly half the time of the baseline, thanks to improved sample efficiency. We emphasize that these timings should be interpreted as approximate due to uncontrolled variability (e.g., concurrent processes, shared hardware), but they nonetheless provide a clear picture of the relative computational footprint.

In Figure 3, it is finally possible to view some examples of decoded sentences, selected to give an idea of the typical errors made by the models. The individual contributions of each architectural component, instead, are validated through a comprehensive ablation study presented in Appendix B. We also provide in Appendix C a study investigating cross-subject generalization in our model.

---

[5]https://www.kaggle.com/competitions/brain-to-text-25/overview

## 4 DISCUSSION

Our results provide several key insights into the potential of Whisper-based neural decoders for speech BCIs.

**Robustness of Performance.** A first major finding concerns the stability and consistency of performance. When trained in the cross-dataset regime, our model achieves WERs that match or surpass the Willett-style baseline across all scenarios, with particularly strong improvements in decoding without 5-gram post-processing. Even the model trained exclusively on the Card dataset outperforms the baseline in this lightweight setup, though it falls behind once heavy n-gram rescoring is applied. This is likely due to Whisper's integrated decoder, which acts as a weak language model and compensates for one of the limitations of CTC-based approaches that depend heavily on external language models. One might question the fairness of the comparison, given that the cross-dataset model is exposed to a larger training corpus. However, our model was trained with substantially fewer updates–only about one quarter of those used for the baseline with the same batch size (Table 2)–and still achieves higher accuracy. This underscores both the sample efficiency of Whisper-based architectures and their ability to leverage prior linguistic knowledge. As expected, absolute WERs are lowest with 5-gram rescoring, since this model was trained on an extremely large corpus of 634M sentences and is well aligned with the short average sentence lengths in our tasks, diminishing the advantage of attention-based models in capturing long-range dependencies.

**Benefits of Cross-Dataset Training.** A second, and perhaps more surprising, message is that cross-dataset training not only supports generalization across recording conditions but also improves performance on the original dataset itself, even without additional fine-tuning. This suggests that the architecture is particularly well suited to benefit from exposure to larger and more diverse data distributions. To our knowledge, this is the first demonstration of cross-dataset advantages in the context of intracortical neural speech decoding. This finding has important implications for future research, where the availability of broader datasets may enable more generalizable and data-efficient BCIs.

**Computational and Practical Considerations.** The third key message relates to the computational trade-offs of different decoding strategies. While 5-gram rescoring yields the best WERs, it requires ∼300GB of RAM and substantially longer inference times, making it unsuitable for deployment on common hardware. This constraint is especially problematic for BCIs, where privacy concerns render cloud-based processing undesirable. In contrast, our Whisper-based model requires less than 2GB of VRAM for inference and achieves sub-100ms decoding latency without n-gram rescoring, making it suitable for local, on-device decoding. This property broadens the potential use of neural speech decoders and aligns with the need for privacy-preserving, real-time communication solutions.

**Limitations.** Despite these promising results, several limitations remain. First, there is a fundamental scale mismatch between Whisper's pretraining corpus (680k hours of audio) and the ∼40 hours of intracortical recordings available across two participants. This disparity constrains the extent to which Whisper's large capacity can be adapted to neural inputs and likely limits the specialization of its early layers. Second, neural recordings remain highly non-stationary. While hierarchical day normalization improved stability, performance still degrades with temporal distance between training and evaluation sessions. More robust solutions, such as continual learning or lightweight calibration strategies, will be necessary to ensure long-term deployment. Finally, although our decoder demonstrates strong sample efficiency, its reliance on high-quality invasive data limits generalizability to broader, non-invasive applications.

**Ethical and Privacy Considerations.** The ethical implications of neural speech decoding are profound. Thoughts, and by extension neural data, represent some of the most private information of a human being. While recent advances in BCI technologies have shown extraordinary promise in restoring communication for people with severe impairments, they also open the door to potential misuse. It is therefore crucial to begin considering neurorights from a regulatory perspective (Yuste et al., 2017). One specific concern is the possibility of decoding neural activity against the user's will. Current evidence suggests that successful decoding requires user collaboration and can be

disrupted by deliberate attentional strategies, such as mental counting, naming unrelated words, or engaging in inner monologue, which interfere with the inference process (Tang et al., 2023). More recently, Kunz et al. (2025) demonstrated the feasibility of "mental passwords", where decoders are trained to initiate output only after detecting a user-specific cognitive trigger. Such safeguards, empowering the user with intentional control, are essential to ensure that decoding systems remain transparent, safe, and under volitional use.

A second ethical dimension concerns the reliability of decoded outputs. Speech restoration is central to quality of life, yet decoding systems are inherently imperfect. Errors may result not only in communication breakdowns but also in biased or unintended outputs that fail to reflect the user's true intentions. Future research should explore strategies to mitigate these risks, such as adaptive filters that suppress low-confidence outputs or hybrid systems that allow switching between fast, moderately accurate decoding and slower, high-accuracy decoding depending on user needs. These mechanisms could help prevent the inadvertent overriding of user intent by model biases.

Finally, the sensitivity of neural data raises critical issues around data storage and processing. Cloud-based inference pipelines pose unacceptable risks, as outsourcing neural data could expose individuals to surveillance or misuse. A key requirement for ethical deployment is therefore the ability to run inference locally, on affordable and resource-constrained devices, without the need to transmit raw neural data externally. Our work contributes to this goal by demonstrating a lightweight architecture that achieves real-time decoding under minimal hardware requirements.

Taken together, these considerations highlight that technical innovation in BCI must proceed hand in hand with safeguards that preserve privacy, autonomy, and the authenticity of user communication.

**Future Directions.** Looking ahead, one promising direction is to establish tighter links between neural activity and the acoustic domain. While our current system predicts phonemes as an intermediate representation, future work may move toward end-to-end neural-to-speech models. Achieving this will require innovative solutions to the lack of intelligible audio from participants with severe impairments, potentially leveraging synthetic data or self-supervised representations such as Hu-BERT (Hsu et al., 2021) and wav2vec 2.0 (Baevski et al., 2020). Diffusion-based speech synthesis methods also hold promise for producing fluent and personalized outputs directly from neural activity.

**Toward Foundation Models for Neural Decoding.** Ultimately, this study represents a step toward foundation models for neural speech decoding: large pretrained systems capable of generalizing across subjects, sessions, and tasks with minimal adaptation. By combining inductive biases such as local attention and day normalization with the powerful priors of pretrained ASR architectures, we move closer to a scalable framework for neural BCIs. Bridging the gap between data-hungry large models and the inherently low-data, high-variability conditions of clinical BCIs remains an open challenge. Nevertheless, our findings demonstrate that Whisper-based architectures provide a promising pathway toward this goal, enabling robust, efficient, and privacy-preserving speech restoration.

## 5    CONCLUSIONS

In this work, we introduced a neural speech decoder that leverages the representational power of large-scale pretrained ASR models to translate intracortical recordings into text. By directly projecting neural features into Whisper's encoder and training under a hybrid phoneme–token objective, our system achieves robust performance while maintaining low computational requirements. Across experiments, we showed that cross-dataset training not only enhances generalization but also improves performance on the original dataset without additional fine-tuning, underscoring the scalability of this approach. Moreover, the lightweight inference footprint of our decoder makes it suitable for local deployment, an essential property for privacy-preserving BCIs. While limitations remain–particularly the scarcity and non-stationarity of neural data–our findings demonstrate that Whisper-based architectures offer a scalable and efficient pathway toward real-time speech neuro-prostheses. Taken together, these results represent a concrete step toward foundation models for neural decoding, capable of bridging the gap between high-performing ASR systems and the stringent requirements of clinical BCI applications.

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

## ETHICS STATEMENT

This study makes exclusive use of publicly available intracortical data (Willett et al., 2023; Card et al., 2024), which were collected with informed consent under protocols approved by the respective

institutional review boards. Potential risks of brain decoding research, such as privacy concerns and possible misuse, are acknowledged. The work is intended solely to advance scientific understanding and should not be used for individual level prediction or surveillance.

## REPRODUCIBILITY STATEMENT

Preprocessing pipelines, model architecture, training objectives, and hyperparameters are detailed in the manuscript. All experiments can be reproduced with the scripts provided as a zipped repository in Supplementary Materials. We highlight that large language models (LLMs) were used exclusively for textual editing and polish writing.

## A   TRAINING DETAILS

In this section, we provide details of the training pipeline and hyperparameters used in our experiments to facilitate reproducibility. Our models are based on the Hugging Face implementation of openai/whisper-tiny.en (hidden dimension 384, i.e., $D = 384$). Pretrained weights were retained for all modules that were not modified.

Training was performed with the Adam optimizer (Kingma & Ba, 2015) for either 20k steps (Card-only training) or 26.4k steps (cross-dataset training), using a batch size of 64. We employed a cosine learning rate scheduler that decayed the learning rate from $1 \times 10^{-3}$ to $1 \times 10^{-5}$, with a weight decay of $1 \times 10^{-5}$. The Whisper decoder was kept frozen throughout training. Checkpoints were selected based on the best validation PER or WER (used respectively for post-processing or greedy decoding).

For the convolutional layers in the embedder, we used kernel sizes of 7 and 3, with a stride of 2 for the second convolutional layer. A dropout rate of 0.4 was applied during training. The phoneme prediction head was attached to the third encoder layer ($L_p = 3$), and the window size for the windowed attention mechanism was set to $w = 14$.

For the Card dataset, we employed a rank-16 day-specific projection, while for the Willett dataset we used full-rank projection matrices. Major hyperparameters were tuned via random search. Additional implementation details are available in the shared code.

## B   ABLATION STUDY

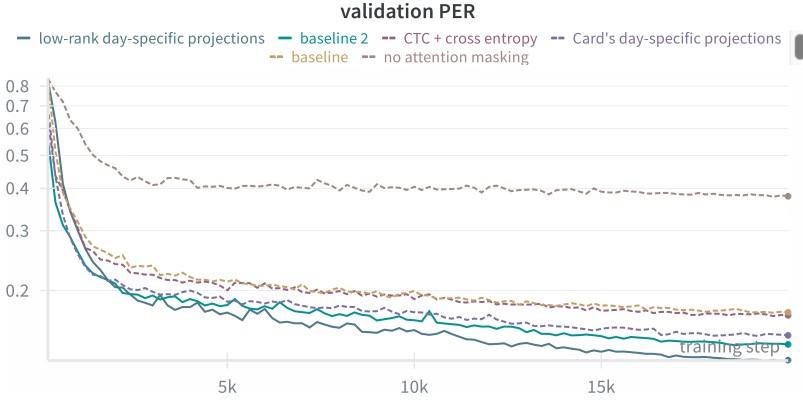

Figure 4: Ablation study results showing validation PER over training steps for different model configurations. Dashed lines represent the first phase of ablation, where individual components are added to or removed from a minimal baseline (no day-specific projections, CTC loss only). Solid lines represent the second phase, comparing models with all beneficial components while varying the rank structure of day-specific transformations.

To systematically evaluate the contribution of each proposed component, we conducted a comprehensive ablation study on the Card dataset, examining how individual architectural choices affect model performance. Results in terms of validation PER are reported in Figure 4.

The ablation study is structured in two phases, each addressing different aspects of our architectural contributions. In the first phase (represented by dashed lines in Figure 4), we analyze the impact of key model components relative to a minimal baseline. This baseline model was trained on Card data using identical hyperparameters to those employed in the main experiments, with two critical modifications: no day-specific projections were applied, and the cross-entropy loss component was disabled, leaving only the CTC phoneme prediction objective active. From this baseline configuration, we observe several important trends. Disabling windowed attention masking in the early encoder layers leads to a substantial degradation in PER, confirming our hypothesis that articulatory dynamics benefit from local temporal context rather than global attention patterns. This finding aligns with the interpretability studies of Whisper's encoder layers, which reveal strongly localized receptive fields in early phoneme-selective representations. Conversely, incorporating the cross-entropy loss component for token-level supervision yields a measurable improvement in phoneme prediction accuracy, demonstrating that the dual objective successfully encourages the encoder to learn representations that are both phoneme-discriminative and linguistically coherent. The addition of day-specific projections further enhances performance, highlighting the importance of addressing session-to-session variability in neural recordings.

The second phase of the ablation study (represented by solid lines in Figure 4) focuses specifically on the impact of low-rank factorization in the day-specific transformation layers. Here, we compare models that incorporate all beneficial components identified in the first phase while varying the rank structure of the day-specific projections. The results demonstrate that employing low-rank day-specific projections achieves the best overall performance in this configuration. This finding supports our design choice to use the factorized form, which not only reduces computational complexity and prevents overfitting to individual sessions but also appears to provide a more effective regularization mechanism than full-rank transformations.

These ablation results provide empirical validation for each of our proposed architectural modifications and demonstrate that the performance gains achieved by our full model arise from the synergistic combination of multiple complementary innovations rather than from any single dominant factor.

## C CROSS-SUBJECT GENERALIZATION

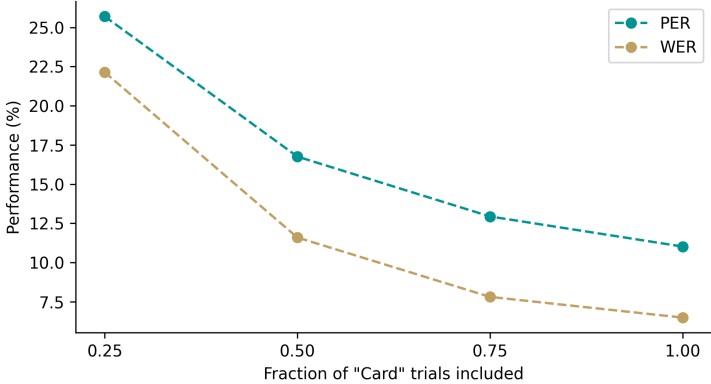

Figure 5: Cross-subject generalization with varying amounts of target-subject data. Models were trained on the complete Willett dataset combined with increasing fractions of the Card training set, then evaluated on Card's test split. Performance is reported in terms of PER and WER.

An important question not addressed in the main body of the article is whether the model can decode neural signals from subjects never encountered during training, thereby achieving cross-subject generalization. By design, our model cannot generalize zero-shot to unseen subjects because different

participants typically have varying numbers of implanted electrodes located in non-corresponding brain regions. Even if subjects had identical electrode counts and placements, substantial inter-subject variability would remain: channel $n$ in subject $X$ may not carry the same neural information as channel $n$ in subject $Y$. We partially address this challenge through the month- and day-specific projection layers, which map neural signals from different subjects into a shared representational space. However, optimizing these projections requires at least a minimal amount of recording data from the target subject. Thus, while zero-shot cross-subject generalization is not feasible, adaptation to new subjects becomes possible given a modest amount of target-subject data.

To investigate the efficacy and data efficiency of this adaptation process, we conducted an experiment in which models were trained on the full Willett dataset combined with increasing proportions of the Card dataset, then evaluated on Card's held-out test split. Training followed the same pipeline and hyperparameters used throughout our main experiments. Results are shown in Figure 5. The findings reveal that acceptable performance can be achieved using as little as 20% of the target subject's training data. To approach the psychologically significant threshold of 10% WER–beyond which decoders are typically considered viable for practical deployment–at least 50% of the original training set is required. Beyond this point, performance improves gradually and approaches that of the full training regime, with diminishing marginal gains. These results demonstrate that our model can be effectively adapted to new subjects with limited data availability and suggest that performance could be further enhanced through more sophisticated adaptation techniques such as subject-specific fine-tuning.

