# OpenReview forum: "Towards real-time BCI for speech with Whisper-based decoding of neural activity"
_ICLR.cc/2026/Conference — Submitted to ICLR 2026_

### Official Review · Reviewer_4krM · 2025-10-27

**Soundness:** 2
**Presentation:** 3
**Contribution:** 2
**Rating:** 2
**Confidence:** 4

**Summary:**

The paper introduces Whisper-BCI, a neural speech decoder that maps MEA recordings into a pretrained ASR model (Whisper) to improve robustness and cross-subject generalization. Evaluations are performed to show effectiveness.

**Strengths:**

* a) This paper is well organized and easy to follow.

**Weaknesses:**

**(a) Limited novelty**
The proposed method builds directly on the  Whisper architecture. The proposed architecture still follows the CTC loss training with beam search for inference, with an additional CE loss for as the token-level loss being the only difference from Willett et al. As such, the work primarily represents an engineering integration of existing approaches rather than introducing new insights from either the neuroscience or the machine learning perspective.

**(b) Incremental improvement**
The reported performance in Table 1 is only marginally better than the baseline models used in prior work, and these gains appear modest relative to the complexity added by the proposed approach. This raises questions about the practical advantage of the method over simpler existing techniques.

**(c) Insufficient Ablation Study**
Ablation should at least be conducted to demonstrate the effectiveness of the proposed approach compared to the base model proposed by Willett et al. to show solidness.

**Questions:**

N/A

---

> ### Author Response · Authors · 2025-11-25
>
> We thank the reviewer for raising these concerns, as addressing them provides an opportunity to clarify the scope and significance of our contributions. Regarding novelty, we believe our work represents one of the first successful applications of transformer-based architectures to speech decoding from intracortical MEA recordings and the first to leverage the close relationship between this task and ASR by adapting a pretrained ASR model. This constitutes a major departure from the GRU-based framework of Willett et al. Beyond the architectural shift, we introduce several domain-specific innovations. The month- and day-specific projection layers effectively mitigate the non-stationarity of neural signals, with the low-rank factorization of day-specific projections enabling efficient use of the limited number of trials available per recording session. The use of windowed self-attention in early encoder layers, motivated by Whisper interpretability studies and neuroscientific considerations of articulatory dynamics, further contributes to model performance. The multitask learning setup overcomes the conditional independence assumption inherent in CTC loss and incorporates distributional priors from natural language into the learned representations. Finally, and importantly, we emphasize the originality and significance of our cross-dataset experiments—to our knowledge, the first in the MEA-based speech decoding literature. These experiments reveal that training on multiple subjects improves decoding performance on individual subjects without requiring additional fine-tuning, a finding with substantial implications for the development of more generalizable decoding systems.
>
> Regarding incremental improvements, we partially agree with the reviewer concerning results obtained with 5-gram post-processing. However, we emphasize that the results obtained via greedy decoding from our word head are highly competitive, achieving approximately a 50% relative reduction in WER compared to the corresponding baseline. This advantage is even more pronounced when considering computational profiles: our greedy decoding path saves approximately 12GB of VRAM and achieves inference times 15x faster than the baseline, enabling deployment on resource-constrained devices such as smartphones.
>
> Finally, we have conducted an ablation study in Appendix B, where we systematically evaluate the contribution of each proposed component, including the multitask learning framework, low-rank month- and day-specific projections, and windowed attention mechanisms.

---

### Official Review · Reviewer_SXv4 · 2025-10-31

**Soundness:** 3
**Presentation:** 3
**Contribution:** 2
**Rating:** 4
**Confidence:** 4

**Summary:**

The authors propose fine-tuning a Whisper model on intracranial MEA recordings to decode speech from the brain. They train their model to provide two decoding pathways: (1) where phoneme representations are rescored via WSFT, and (2) a more efficient pathway that leverages the Whisper decoder as an implicit language model and uses a small beam search to produce fast transcriptions. The authors run experiments where they jointly train with the Willett and Card data and show cross-subject generalisation and comparable performance to the current state-of-the-art on the respective datasets.

**Strengths:**

- Promising idea to combine both phoneme representations with word token predictions as they likely leverage complementary neural signals
- Aggregating all publicly available MEA datasets is a sensible step for scaling intracranial decoding
- Hierarchical normalization via month- and day-specific projections to account for representation drift is well-motivated
- Lightweight ASR approach by skipping the WSFT is sensible as the field moves towards real-time intracranial BCIs

**Weaknesses:**

- Improvements over Card and Willett are minor, if any, and are difficult to assess without error bars or a sense of statistical significance. Is this possible to gather via the competition platform?
- I would like to see a more robust analysis of subject generalisation as this could be an important aspect of this work. How does zero-shot subject transfer look? Does training on X% of Willett yield the same gain on Card as training on X% of Card?
- The authors have not made comparisons to the relevant baselines they discuss (e.g. LISA) and others described in the original Willett competition reflections paper [A]
- Minor: Line 105; NeuSpeech did not re-train Whisper on MEG recordings but rather fine-tuned the model with MEG.

I am open to raising my score if the authors can satisfactorily address the above weaknesses.

[A] Willett, F.R., Li, J., Le, T., Fan, C., Chen, M., Shlizerman, E., Chen, Y., Zheng, X., Okubo, T.S., Benster, T. and Lee, H.D., 2024. Brain-to-Text Benchmark'24: Lessons Learned. arXiv preprint arXiv:2412.17227.

**Questions:**

Questions
- Line 124-127: I understand how month- and day-specific projections help with representation drift across time, but how does it improve generalisation “across subjects”?
- Line 239-240: Although local window attention is a smart choice for computational efficiency, articulatory dynamics are not the only factor that may influence phoneme prediction. Low-frequency long-range semantic signals could inform and improve these predictions, too. Did the authors try training with full attention?

---

> ### Author Response · Authors · 2025-11-25
>
> We are pleased that the reviewer appreciated our multitask learning framework, the effort invested in aggregating multiple MEA datasets, the introduction of low-rank day-specific projections to mitigate non-stationarity, and the advantages of our end-to-end decoding branch.
>
> Turning to the identified weaknesses: as the reviewer correctly anticipated, we are currently unable to provide error bars or conduct rigorous statistical significance tests, as the competition platform imposes strict limits on the number of allowed submissions.
>
> Regarding the proposed generalization analyses, we clarify that our model was not designed for zero-shot cross-subject generalization. Even when subjects are implanted with the same type and number of electrodes in approximately corresponding cortical regions, the functional mapping between individual channels across subjects remains unclear or entirely non-identifiable. We address this challenge through month- and day-specific projection layers, which require at least some target-subject data to be optimized. As an alternative, we provide a cross-subject generalization analysis with limited data in Appendix C, demonstrating that our model can successfully adapt to new subjects using only a modest fraction of their recording sessions.
>
> Regarding the addition of baselines mentioned by the reviewer: LISA relies on the same GRU-based phoneme decoder introduced by Willett et al. and subsequently employed by Card et al., which already serves as our primary baseline. As for the Linderman Lab's and DCoND decoders, we are not aware of any publicly released code, and implementing these models from scratch—including training pipelines and full model training—is not feasible within the revision timeframe.
>
> We regret that we were unable to address all of the reviewer's requests, which we agree are both interesting and appropriate, and we hope to have clearly and convincingly explained the rationale behind our choices. Finally, we thank the reviewer for the careful evaluation that identified the error on line 105, which has been promptly corrected.
>
> We provide below our responses to the reviewer's questions:
> - The month- and day-specific projections are inherently subject-specific, as each recording session is uniquely associated with a particular participant. Once optimized during training, these learnable projection layers map subject-specific neural features into a shared representational space that is common across subjects. Different participants are typically implanted with varying numbers of electrodes in non-corresponding cortical locations, making direct feature alignment impossible. Our projection layers resolve this by transforming heterogeneous channel configurations into a fixed-dimensional "virtual" feature space, where each virtual feature can be interpreted as a learned combination of the original recording channels.
> - We agree with the reviewer that long-range semantic context can in principle inform phoneme predictions. However, our architectural design explicitly accounts for this consideration through the structure of the encoder. Windowed attention is applied only to the first layers, which are responsible for constructing phoneme-selective representations. The remaining encoder layers, which operate with full global attention, are tasked with integrating broader linguistic and semantic context. To empirically validate this design choice, we conducted experiments with full global attention across all encoder layers. Results are presented in the ablation study in Appendix B and confirm that models trained with windowed attention in the early layers consistently outperform those using full attention throughout. This finding seems to support our hypothesis that phoneme-level representations benefit from local temporal context.

---

> > ### Comment · Reviewer_SXv4 · 2025-11-25
> >
> > Thank you for your response and for clarifying the difficulty of testing the baselines in the competition paper.
> >
> > Regarding error bars, the authors could run more seeds and evaluate on the test set of the Willett data rather than on the held out competition set.
> >
> > With regard to the generalisation analysis, I am not proposing to do zero-shot cross-subject generalisation. My suggestion is to understand the extent to which the model leverages other datasets in training to achieve cross-subject generalisation. Given a fixed number of training samples, the authors could train with 90% dataset A + 10% dataset B and evaluate generalisation to dataset B vs training on 100% dataset B. This is a minor suggestion to help understand the contribution or efficiency of cross-subject generalisation.
> >
> > Given the limited novelty and incremental improvement, unfortunately, I am not able to recommend acceptance without further evidence.

---

### Official Review · Reviewer_1X1X · 2025-11-02

**Soundness:** 2
**Presentation:** 2
**Contribution:** 2
**Rating:** 2
**Confidence:** 5

**Summary:**

This paper proposes Whisper-BCI, a brain–computer interface system that decodes speech directly from Brain-to-Text 2025 recordings using a modified version of the Whisper automatic speech recognition (ASR) model. The authors project neural features into Whisper’s encoder and optimize it with a hybrid loss that combines phoneme-level CTC loss and token-level cross-entropy.

**Strengths:**

The idea of utilizing some proven models in speech to text generation and migrate these models into brain signal to text generation makes sense. In this area, we've observed quite a bit papers following the same phylosophy.

**Weaknesses:**

- The experiments are extremely limited, where only one set of experiment reported and the performance is even not close to baseline. Is this results just to illustrate whisper model could reach slightly worse results compared to the baseline? Also, in the original brain-to-text 2025 repo, there are already some wav2vec/whisper similar structures.

- Utilizing whisper model into brain to text decoding is not new as well. What is the difference between this paper and NeuSpeech paper using MEG data? Merely the data sampling rate and the shape difference won't bring too much novelty.

**Questions:**

-

---

> ### Author Response · Authors · 2025-11-25
>
> We thank the reviewer for recognizing the value of leveraging advances in ASR for neural speech decoding. Regarding the perceived weaknesses, we respectfully note that PER is the only metric on which our best model underperforms relative to the baseline. Moreover, while PER provides insight into phoneme-level accuracy, we argue that it is not the most relevant metric for practical communication BCIs, as human conversational partners are accustomed to reading words rather than phoneme sequences. For this reason, we emphasize WER as the primary evaluation metric. By this measure, our best model matches or outperforms the baseline across all scenarios, with substantial margins when 5-gram post-processing is not applied.
>
> Regarding the official Brain-to-Text 2025 repository, we are unfortunately unable to locate references to wav2vec or Whisper architectures.
>
> Compared to the NeuSpeech paper, our work introduces several key contributions that were critical to achieving the reported performance. First, we adopt a multitask learning framework in which the standard cross-entropy loss is complemented by a CTC loss for phoneme prediction. This design was motivated by recent interpretability studies showing that Whisper's encoder constructs phoneme-selective representations in early layers and word-level representations in later layers; accordingly, we attach the phoneme prediction head at the third encoder layer. Second, we address the non-stationarity inherent in intracortical MEA recordings through month-specific and low-rank day-specific projection layers, which map neural signals into a shared, session-invariant space. Third, we employ windowed self-attention in the encoder layers responsible for phoneme representations, motivated by the hypothesis that cortical neurons encode the phoneme of interest along with a limited temporal context. All of these contributions are detailed in Section 2.2 of the manuscript.

---

### Meta-Review · Area_Chair_XpUx · 2025-12-26

**Summary:**

The paper presents a technically sound approach that adapts an open-source ASR model to the task of decoding speech from intracortical multi-electrode array recordings. The authors introduce several domain‑specific engineering improvements and demonstrate modest gains in WER, especially when cross‑subject data are aggregated and when post‑processing is applied. Authors argue that their approach is especially useful in constrained compute experiments.

However, the contribution is incremental in the context of the BCI community, similar ideas appear in NeuSpeech and other ASR‑inspired BCIs. The empirical evaluation is narrow (one primary dataset, limited analysis), and the reported gains over established baselines are small or only evident after heavy post‑processing, and limited to certain compute budgets. Authors have valid reasons for not being able to provide deeper analysis & ablations, however this does not alleviate the reviewers' concerns.

**Reviewer Concerns:**

Novelty is a bit contested contested: authors point to their use of transformers and several domain specific tricks, but reviewers don't think these are valid and significant innovations

Evaluations & Ablations: reviewers still expect broader and deeper evaluations that demonstrate that the proposed approach outperforms baselines.

Efficiency: one of the authors' main arguments is the efficiency of the proposed approach. However this is not much discussed so far and still outstanding.

**Reviewer Scores:**

I don't think reviewers would have substantially raised their scores, see SXv4's remarks.

---

### Decision · Program_Chairs · 2026-01-26

Reject